# “We Want to See Youth That Would Be Better People Than Us”: A Case Report on Addressing Adolescent Substance Use in Rural South Africa

**DOI:** 10.3390/ijerph20043493

**Published:** 2023-02-16

**Authors:** Ifeolu David, Lisa Wegner, Wilson Majee

**Affiliations:** 1Department of Health and Rehabilitation Science, School of Health Professions, University of Missouri, Columbia, MO 65211, USA; 2Department of Occupational Therapy, Faculty of Community & Health Sciences, University of the Western Cape, Cape Town 7535, South Africa; 3Department of Health Sciences and Public Health, University of Missouri, Columbia, MO 65211, USA

**Keywords:** substance use, coalition, women, adolescents, rural, South Africa

## Abstract

Research suggests that substance use, particularly among adolescents may lead to increased illegal activities as well as physical and social health consequences. Around the world, communities, overburdened with adolescent and youth substance use, are finding ways to address this public health threat. Using a case study based on a focus group discussion with nine founding members, this paper reports on the case of Sibanye—a rural community coalition whose mission is to reduce the burden of adolescent substance use on families in rural South Africa. The focus group discussions were audio-recorded, transcribed verbatim, and analyzed using Nvivo 12. This work highlights the power of an engaged collective effort in addressing key community issues, even in rural areas of emerging economies where health and community infrastructure may be limited. The Sibanye coalition taps into its collective wealth of community knowledge to provide social contributions and aesthetics to help adolescents abstain from substance use and sexual risk behaviors. These activities offer adolescents safe places to meet each other, health education, and the ability to structure their free time meaningfully. Engaging community residents, particularly disadvantaged groups, should be central to any local and national strategies for promoting health and well-being.

## 1. Introduction

Today, communities around the world face a host of problems, including joblessness, poverty, a growing population of disengaged youth, and substance abuse [1,2,3]. Substance use (used broadly to include alcohol, tobacco, and illicit drugs) by adolescents remains a significant public health concern imposing an enormous burden on individuals, families, and communities, hence diminishing quality of life [4,5,6]. Substance use behaviors formed early in life can have individual and societal consequences that span a lifetime, including poor school attendance and academic performance, high rates of school dropout, criminal activities, loss of employment, unproductivity, weak prospects for marriage, health-related problems, and increased societal costs [7,8,9,10,11]. Young people who misuse alcohol and drugs experience more medical symptoms, such as appetite changes, weight loss, headaches, sleep disturbance, and depression, than their counterparts, with negative effects on learning and development [12,13,14]. Rates of mental, neurological, and substance use disorders are highest during adolescence and early adulthood [15,16].

In Africa, recent research highlighted a high prevalence of substance use among young people when compared to the general population, with associated physical and psychosocial problems such as fighting, vandalism, theft, unprotected sex behaviors, personal injury, medical problems, and impaired relationships with family and friends [17,18,19]. In South Africa, commonly used substances include alcohol, methamphetamine, cannabis, and nyaope (a mixture of antiretroviral drugs, low-grade heroin, cannabis, and other materials such as rat poison added as bulking agents) [20]. Despite national efforts to combat substance use by reducing the supply of illicit substances [21], the enforcement of such legislation has been weak in resource-limited settings, thus resulting in less controlled marketing practices [22]. Further, job losses and school closures associated with the COVID-19 pandemic, amplified substance use in Africa, which was already on the rise prior to the pandemic [23].

While there is a lot of research on substance use and tobacco control, including how to employ advanced analytical techniques, such as machine learning, to inform policy decision making around tobacco control [24,25], little has been done to showcase the work of community-based organizations in resource-limited rural communities in addressing substance use among adolescents. Using a case study based on a focus group discussion with nine founding members, this paper reports on the case of Sibanye—a rural community coalition whose mission is to reduce the burden of adolescent substance use on families in rural South Africa. The characteristics, functions, challenges, and impacts of Sibanye are discussed in this paper.

## 2. Methods

### 2.1. Study Setting and Participant Selection

The Sibanye group is a community coalition formed by 10 women to address the problem of youth substance use and its consequences in their community. Derived from a Zulu word meaning “We are one”, the organization was founded in 2017, following a spike in substance use among adolescents and youth, increased cases of violence, vandalism, and school drop-out rates. The coalition is located in and serves the Sea Vista community. Sea Vista is a community in the Kouga Local Municipality in rural Eastern Cape. One of the researchers has a working relationship with the Kouga Municipality, with whom we partnered to conduct the larger study. The case study was conducted as a supplementary project for a larger study that explored risk behaviors among rural South African youth. Findings from the larger study are published elsewhere [2,3,26]. During data collection for the larger project on adolescent risk behaviors, it came to our attention that there was a group of women in one of the poorest communities in the municipality who were working to address substance use. We then contacted one of the women and invited them to participate in our study. Of the ten founding members of Sibanye, nine consented to participate. Five of the members were married, four were single, with an average of two children under the age of 18 years. The women were, on average, 41 years of age, had lived in the community for 9–35 years, and were fluent in a range of languages, including English, Afrikaans, Xhosa, SeSotho, Zulu, and Shona (Table 1).

### 2.2. Data Collection

The study used an exploratory, descriptive, contextual qualitative design to explore perceptions and experiences of Sibanye’s members on the role and function of the coalition in addressing substance use among adolescents. According to Creswell and Poth [27], to study a problem, qualitative researchers use an emerging qualitative approach to inquiry and the collection of data in a natural setting sensitive to the people and places under study. Burns and Grove [28] define exploratory research as research conducted to gain new insights, discover new ideas and/or increase knowledge of a phenomenon. A contextual design refers to and focuses on specific events in “naturalistic settings” [28]. Three methods of data collection were used: (a) a brief questionnaire to obtain demographic information; (b) a semi-structured interview protocol used in a focus group discussion (FGD) with all nine participants; and (c) a follow-up interview with the first founding member of Sibanye. Two researchers (WM and LW) developed the interview protocol from a review of the literature on vulnerable youth risk behaviors [29,30] and based on concerns raised by youth in the larger study. The interview started with general questions such as, “Please describe what life is like for young people living in this area?” and, “What do you think are good things [for adolescents] living in this community?” Probing questions were used to obtain details. The follow-up interview enabled more in-depth information to be obtained and allowed clarification of specific issues that emerged during the focus group.

Before the focus group, participants were debriefed on the purpose of the study. Prior to participation, all participants were asked to sign a consent form that included permission to audiotape the entire session. During the FGD, the two researchers (WM and LW) acted as facilitators and observers, asking questions and taking notes. The focus group lasted two hours and was conducted in the living room of one of the researchers’ apartment in Sea Vista. Participants received no financial compensation but were offered refreshments at the end of the focus group. The two researchers (WM and LW) held a debriefing session to discuss their own reactions, observations about the group, climate in the room, major themes that arose, and areas of concern. The follow up interview with the main founding member took place one week later at a coffee shop in the community.

### 2.3. Data Analysis

The FGD recording was transcribed verbatim. Data analysis was carried out in phases as informed by consensual qualitative research (CQR) [31], which has been proven to be useful and suitable for focus groups [32]. First, two researchers (ID, WM) engaged in a process of immersion in, and familiarization with, the transcripts [33] and established preliminary domains. Second, the two researchers compared their domains, refined them and agreed on final domains. Third, the researchers returned to the transcripts and independently developed core ideas [31], identified patterns and associations within the data set [34] and corresponding transcript quotes to support the development of the core ideas. In the final phase, the third author (LW) conducted an audit to preserve the integrity of the data. Although the auditor identified additional passage quotes that exemplified the core ideas, no major modification was made to the other two researchers’ work. One author (ID) drafted the manuscript and the other two (WM, LW) reviewed.

All three authors were of African descent. ID is an African-born male graduate student with training in medicine, and currently studying for a PhD in Health and Rehabilitation Sciences. LW is white South African female researcher with expertise in youth leisure and boredom research using qualitative methods. WM is an African-born male researcher with expertise in youth engagement research applying qualitative research.

## 3. Findings: The Sibanye Case

Over the years, Sibanye has emerged as a major resource for vulnerable adolescents and their families providing after-school education (reading) as well as leisure activities including netball, reading, and sewing. Sibanye uses a multi-systems adolescent empowerment approach deeply rooted in constructs of the socioecological model, including interventions at the individual, community, and societal levels to address substance use and the associated negative impact on quality of life.

Sibanye members united towards a shared vision of a substance-free community in their quest to achieve sustainable social change around the substance-use community.

We are a group of 10 African women, different ages, different religions. We came together because we wanted to fight substance abuse and all the other things that come after it, like crime, teenage pregnancies, and school dropout.

We are passionate about making a change in our community. We want to see youth that would be better people than us and make Sea Vista a better place.

To achieve this goal, they took on different roles within the coalition that enabled them to individually contribute towards, and jointly shape, their shared vision. Although members have different responsibilities, their collective effort captures its meaning in collaboratively creating and realizing a sustainable vision for their community. Sibanye members focused on working with and listening to adolescents to build long-term relationships and develop meaningful solutions to the complex issues they face.

My role is to be a role model. I’m talking to girls, trying to make them better people, aware that life is not as easy as they think and to stay away from the dangers that are happening in our community, especially drugs, crime, and abstain from sex so that we won’t have more teenage pregnancies.

Notable impacts of their efforts within the community include introducing adolescent boys and girls to information about current problems associated with drug abuse, early pregnancy, and sexually transmitted infections. Separate girls’ and boys’ groups met regularly with Sibanye members who provided positive affirmation, education, support, and activities. During such meetings, emphasis was placed upon traits such as respect for each other, as well as self-esteem.

It’s about self-esteem, making them aware of the dangers that are happening in our society with drugs, and crime. Just encouraging them to work hard at school. So that’s the empowering. With the girls, we talk about if you do unprotected sex then you become pregnant or you become sick. So my main thing is to actually tell them to abstain until they are old enough to actually know what to do.

An important component of the boys’ group was teaching them to respect girls. This topic was highlighted as a fundamental problem with older male community members who fail to condemn domestic and sexual abuse inflicted on women and girls.

We teach the boys how to respect the girls because most of the boys don’t respect girls. They must know when girls say no they must accept that.

The women used various activities to attract the adolescents to their groups to use their free time meaningfully, and the empowerment happened during the process of the adolescents’ engagement in the activities.

And another reason why we have these activities—netball, reading, sewing, reading—is to keep them busy. They’ve got so much time for themselves and that is the time that they do wrong things.

A combination of these activities is thought to have improved communication among adolescents and youth in the community, thereby enhancing relations between young boys and girls. Members of the Sibanye group believed that their success in reaching this vulnerable population could be integral in shaping the lives of the younger generation especially in delaying or preventing substance use.

However, several difficulties characterized as systemic and social resource-related constrained their efforts. Systemic challenges were linked with inadequate community infrastructure for education, employment, and health.

Sea Vista is a place where, those employed, work mostly in shops (as cashiers) or houses (as gardeners and house cleaners). Therefore, when it comes to income the income is very low. For a parent to take a child to college or university, those things are not possible.

Another thing that the youth are facing is unemployment. We do not have factories in Sea Vista so the unemployment number is high. That is why you see crime going up and it’s mostly (because of) drugs.

Because of poverty and rurality, most families and youth lack employment, economic, and educational opportunities. Deprived of these opportunities, substance use in Sea Vista is initiated early among adolescents and sustained into their youth and adult years. Poverty remained a major theme, viewed as both a precursor and consequence of systemic injustices perpetuated over the years. As a result, Sibanye members are constrained in what they can do, especially given their rurality and continued lack of support from the government.

What I would say about children in Sea Vista is some are privileged, some are not. (…) it is 10% or even 5% that is privileged, because we have parents that are not working at all. There is no income.

Some companies say we want a person with five years’ experience. How do you expect a person from Sea Vista to have five years of experience?

The participants perceived unemployment to result in crime and substance use.

Education poses several challenges. There is only one public primary school in Sea Vista, which is overcrowded. There is no high school and adolescents have to travel a long distance to get to the high school in the neighboring town.

Moreover, another problem, we do not have a high school in Sea Vista. Like now, the problem that we are facing is we have children that have to go to Humansdorp [neighboring town about 6 miles away] for the high school.

Due to resource limitations in rural communities like Sea Vista, schools fail to attract enough qualified teachers. Some schools do not offer science, technology, engineering, and mathematics (STEM) subjects. In the science and technology-driven world we live in, denying young people the opportunity to study STEM subjects feeds the despair that Sibanye members have about the future of the children. The lack of sufficient social resources was seen as an inherent barrier to community development. Participants talked about resource challenges that impacted the coalition’s operations and performance. A major resource challenge was the lack of safe premises for the coalition to do its work in Sea Vista.

We need a safe place where you can lock everything because we have to put our stuff in there, it must be safe so that no one can go in there and steal because with these drugs they just break in and take everything to sell. 

The participants perceived vandalism and theft to be a consequence of substance use and ascribed this to a lack of connectedness of adolescents to their community.

Vandalism in Sea Vista is happening [firstly] because of drugs. When there is a shop and they realize that there is not enough security they vandalize because they want to steal the stuff to sell. And [secondly], they don’t respect or love their community enough because if you love your community and you see a paper on the ground you pick it up.

Participants reported a lack of sports and recreational facilities which resulted in adolescents having large amounts of free time with no meaningful activities to engage in.

It is because our children or the parents do not have the relevant resources. That is why all of them end up just sitting at home and doing nothing.

Additionally, some participants felt that there were community members who were not supportive of the efforts of Sibanye. This lack of community support constrained Sibanye’s efforts towards building a pool of volunteers the coalition could draw on.

In summary, the participants perceived that these challenges keep the community trapped in a cycle of limited access to basic services, poor educational outcomes, poor economic prospects, and high substance use. However, in the case of Sibanye, the coalition is tapping into its collective wealth of community knowledge to provide social offerings and aesthetics that can help adolescents abstain from substance use and sexual risk behaviors. These activities offer adolescents safe places to meet each other and the ability to structure their free time meaningfully.

## 4. Discussion

Research has shown evidence of a link between unstructured free time and risk behavior [35]. Sibanye’s socioecological approach to addressing multi-level individual, family, systemic, and contextual factors related to the health of adolescents in Sea Vista is unique in three main ways. First, Sibanye is a demonstration of the adage that “educate a woman, you educate a village, a nation, a generation”—the composition of an all-women team shows women can be agents of change and why involvement in development should be acknowledged and elevated. Second, targeting and empowering adolescents helps young individuals in respecting themselves and others, understanding how “inherent” decisions made based on factors influenced by home and community settings can shape their life-long health outcomes. Third, the coalition is addressing individual factors from a holistic perspective through activities that empower adolescents to resist peer pressure, respect girls (for boys), and stay in school [36].

As a collaborative effort working towards a common goal, Sibanye affords the community the opportunity to combine and leverage resources from multiple and diverse sources such as organizations and business donors and has been successful in achieving a collective purpose that is improving the quality of life of adolescents and their families. At its inception in 2017, Sibanye had a few adolescents (4 girls and 3 boys) but has grown to 24 girls and 20 boys, as of October 2022. Sibanye strives to impact the lives of adolescents by teaching them cleanliness, self-love, loving others, and respect. According to one of the founding members, there has been improved communication and relationships between the adolescents they serve and their family members. Families are doing things together more now than before. “Goodness is infectious”, said one Sibanye member. “We have started to see changes in our adolescents’ friends (who are not participating in Sibanye) as a result of the impact we are making on these teenagers”, she added. Rotary International now supports a counseling professional to provide monthly visits with the adolescents for professional emotional support. Over 75% of the adolescents Sibanye serves are in families with only one parent (due to death, divorce, separation, or other factors), and navigating life is a challenge for them given that these families are also poor.

Some of the biggest challenges the coalition has continued to face include a lack of space for their activities, and a lack of support from the community in terms of volunteers. The Community Center available for use is not reliable as there are frequent scheduling conflicts with other community events such as weddings, funerals, and community meetings. The main source of volunteers has been “adolescents who graduated from the program who feel like giving back to a program that opened their eyes”, stated one Sibanye member.

## 5. Conclusions

Insights gleaned from this case suggest that community coalitions can be an ideal vehicle to engage interested community members in local issues. In many cases, community issues require local solutions, with external players injecting the necessary capital (e.g., human, financial, physical) that local people need to address their concerns. In the case of Sibanye, a group of nine local women is empowering young boys and girls through education and skills, thereby building stronger families and reducing the burden of substance use and risky sexual behaviors among adolescents. One limitation of this study is its strength. It is a study focusing on only one case within a specific context. Another is that it did not solicit experiences of the youth and community members served by the coalition. More research is needed that will explore the unique qualities of community coalitions in a variety of settings, both in rural and urban communities, and understand the factors that make some community coalitions effective.

## Figures and Tables

**Table 1 ijerph-20-03493-t001:** Participant Socio-demographic Information.

Characteristics (*n* = 9)	*n* (%)
Gender	
Male	0 (0)
Female	9(100)
Ethnicity	
Colored (mixed race)	6 (67)
Black	3 (33)
Education level	
Some High School	5
Grade 12	2
College	2
Marital status	
Married	5
Single	4
Children under 18 years	
4	1 (11)
3	2 (22)
2	4 (45)
1	2 (22)
0	0 (0)
Household income (monthly—ZAR)	
<1000	1 (11)
1000–4999	4 (45)
5000–9999	3 (33)
>10,000	1 (11)
	Mean (SD)
Age	41.4 (9.76)
Length of stay in community	20.3 (7.9)

## Data Availability

The data presented in this study are available on request from the senior author: Wilson Majee. The data are not publicly available due to maintaining IRB compliance with study protocols.

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
