# Peer review of "“We Want to See Youth That Would Be Better People Than Us”: A Case Report on Addressing Adolescent Substance Use in Rural South Africa"

_ijerph, 2023, doi:10.3390/ijerph20043493_

Round 1

Reviewer 1 Report

CENTRAL AND GENERAL ISSUES

Summary

This work highlights the power of an engaged collective effort in addressing key community issues, even in rural areas of emerging economies where health and community infrastructure may be limited. I believe that the result adds value to the literature on substance abuse. However, I think there are some important aspects that need to be improved before recommending its publication in International Journal of Environmental Research and Public Health.

Specific Comments

1. The authors talk in the introduction about the need to also know about substance abuse behavior in communities with limited resources. However, these environments are increasingly complex and require advanced analysis techniques, such as Machine Learning or Artificial Intelligence. In this line, the authors must include a paragraph in the introduction that talks about this and, in addition, cite, at least, these reference papers where they talk about it:

A. Suruliandi, T. Idhaya, S. P. Raja (In Press). "Drug Target Interaction Prediction Using Machine Learning Techniques – A Review", International Journal of Interactive Multimedia and Artificial Intelligence, vol. In Press, issue In Press, no. In Press, pp. 1-15. https://doi.org/10.9781/ijimai.2022.11.002

R. Fu, A. Kundu, N. Mitsakakis, T. Elton-Marshall, W. Wang, S. Hill and M.O. Chaiton, “Machine learning applications in tobacco research: a scoping review”. Tobacco Control, vol. 32, no. 1, pp. 99-109, 2023, https://doi.org/10.1136/tobaccocontrol-2020-056438

2. Another important aspect that I do not see in the paper is the effect that marketing restrictions have on substance abuse. Exposure to consumer marketing causes consumption, for example, of tobacco, to increase. The fact that in countries with limited resources there is no legislation to limit marketing actions can make substance abuse higher. Authors should discuss this in the introduction and cite the following paper:

Almeida, A., Galiano, A., Golpe, A. A., & Martín-Álvarez, J. M.(2021). The usefulness of marketing strategies in a regulated market: evidence from the Spanish tobacco market. E&M Economics and Management. 24(2), 171-188. https://doi.org/10.15240/tul/001/2021-2-011

3. In the conclusions I do not see any paragraph showing the limitations of this work. It would be important for the limitations of this paper to be made clear.

Reviewer 2 Report

Adolescent and youth substance use is of public health concern, especially in low-resourced settings and rural environments with fewer access to resources. Community based organizations have promise in engaging communities and youth in helping prevent substance use among this population. This manuscript reports qualitative work from a community based organization in rural South Africa addressing this public health concern. This topic is of public health importance and would be of interest to the readers of IJERPH. I have some major and some minor comments:

Major Comments

1.     My main concern about the article is the results presented and the authors’ tendency to ascribe causal associations between the collective and outcomes in the community. This study was designed not to test whether or not Sibanye actually changed youth behavior, but rather to understand activities from the perspective of the women participating in the group. The author should reconsider how some of this is presented to account for what data they actually have.

2.     To me it would have been more interesting to learn about the women’s experience creating this collective and how they feel youth are interacting and less about their opinions about what youth feel. For example, the paragraph starting with Line 277 is quite interesting and more relevant to what the women can speak to. Additionally, this portion was not mentioned in the results section but talked about in the discussion section. It seems that this should first be described in the results section and then commented on in the discussion section.

Minor Comments

1.     Line 32: There are some broken links to references.

2.     Line 66: It would be best to specify the universities that provided IRB approval.

3.     Line 193: If this is a participant quote it is difficult to delineate from the authors’ text. This problem occurs several times throughout the text.

Round 2

Reviewer 1 Report

The authors have included all suggested enhancements.

Reviewer 2 Report

Reviewer comments sufficiently addressed.